

# Diversity of planktonic fish larvae along a latitudinal gradient in the Eastern Atlantic Ocean estimated through DNA barcodes

Alba Ardura[1], Elvira Morote[2], Marc Kochzius[3] and Eva Garcia-Vazquez[4]

[1] USR 3278-CRIOBE-CNRS-EPHE, Laboratoire d'excellence "CORAIL", Université de Perpignan, Perpignan, France
[2] Biologia Animal, Universidad de Almeria, Almeria, Spain
[3] Department of Marine Biology, Vrije Universiteit Brussel, Brussels, Belgium
[4] Department of Functional Biology, University of Oviedo, Oviedo, Spain

## ABSTRACT

Mid-trophic pelagic fish are essential components of marine ecosystems because they represent the link between plankton and higher predators. Moreover, they are the basis of the most important fisheries resources; for example, in African waters. In this study, we have sampled pelagic fish larvae in the Eastern Atlantic Ocean along a latitudinal gradient between 37°N and 2°S. We have employed Bongo nets for plankton sampling and sorted visually fish and fish larvae. Using the cytochrome oxidase I gene (COI) as a DNA barcode, we have identified 44 OTUs down to species level that correspond to 14 families, with Myctophidae being the most abundant. A few species were cosmopolitan and others latitude-specific, as was expected. The latitudinal pattern of diversity did not exhibit a temperate-tropical cline; instead, it was likely correlated with environmental conditions with a decline in low-oxygen zones. Importantly, gaps and inconsistencies in reference DNA databases impeded accurate identification to the species level of 49% of the individuals. Fish sampled from tropical latitudes and some orders, such as Perciformes, Myctophiformes and Stomiiformes, were largely unidentified due to incomplete references. Some larvae were identified based on morphology and COI analysis for comparing time and costs employed from each methodology. These results suggest the need of reinforcing DNA barcoding reference datasets of Atlantic bathypelagic tropical fish that, as main prey of top predators, are crucial for ecosystem-based management of fisheries resources.

## INTRODUCTION

Bathypelagic and mesopelagic fish are important components of plankton, and are so largely unknown that they have been even called the missing biomass (e.g., *Johnson et al., 2011*). They represent an important part of lower levels in the trophic chain that ends in cetaceans as well as other top predators (e.g., *Bulman, He & Koslow, 2002*; *Walker, Mead*

Corresponding author
Alba Ardura, alarguti@hotmail.com

*& Brownell, 2002*), and are especially sensitive to climate change and other environmental alterations (e.g., *Boeing & Duffy-Anderson, 2008*).

An inventory of the ichthyoplankton community is therefore essential for understanding how the trophic chain and by extension the whole ecosystem function, as well as for timely prediction of changes due to ichthyoplankton alterations. Indeed, such an inventory requires accurate species identification of all fish, fish larvae and eggs present in plankton samples. In some regions DNA is employed for species identification of fish eggs and larvae, and there are species-specific markers useful for this purpose in several fish groups of interest such as megrims and hakes (*Perez et al., 2005*; *Von der Heyden, Lipinski & Matthee, 2007*), cod (*Fox et al., 2005*), horse-mackerel (*Karaiskou et al., 2007*) and others. Other ichthyoplankton communities however are lesser known.

DNA barcoding is a methodology that enables accurate identification of fish species (e.g., *Ward et al., 2005*; *Kochzius, 2009*; *Kochzius et al., 2010*; *Pappalardo et al., 2015*). It was employed to identify blue mackerel eggs and larvae in Australian waters (*Neira & Keane, 2008*), Antarctic fish (*Dettai et al., 2011*; *Belchier & Lawson, 2013*), coral reef ichthyoplankton (*Hubert et al., 2015*) and Amazonian fish (*Ardura et al., 2010*). Barcode of Life projects are considered a promising tool to identify all species, given the expectations of massive decrease in analysis costs (*Edwards et al., 2010*). Today, however, DNA-based methodologies are not routinely employed and the task of ichthyoplankton inventory relies on the taxonomical expertise of zoologists and marine biologists specialized in local pelagic fauna in many regions. A problem is the absence of detailed descriptions of early stages for many species, thus identifying individuals to the species level is frequently impossible. The number of larval fish descriptions for a given region is often inversely related to faunal diversity (*Fahay, 2007*). Following the taxonomic sufficiency approach, *Hernandez et al. (2013)* suggested analyses at the genus level can be a good proxy when examining assemblage diversity.

In this study, we have applied DNA barcoding for identification of fish larvae sampled from a latitudinal gradient in the Eastern Atlantic Ocean. The objective was to identify current needs for the full application of DNA barcoding to the inventory of planktonic fish in large scale surveys. We have employed the methodology proposed by the Fish barcoding of Life initiative (FISH-BOL; *Ward, Hanner & Hebert, 2009*).

## MATERIAL AND METHODS

### Sampling

During the cruise of the RV Polarstern, fish larvae were obtained by means of oblique Bongo hauls (mesh size of 0.5 mm) from a depth of 200 m depth to the surface. One Bongo net sample was designated to barcoding analysis, while the samples from the other Bongo net were used for visual taxonomic identification. After hauling the net back on the deck the cod end of the Bongo net was poured in a bucket and the net was rinsed. To avoid damage to the fish larvae, the cod-end containing larvae for visual taxonomic identification was removed before the net was washed down. The rinsed sample was kept in a separate bucket. The whole sample was examined for fish larvae.

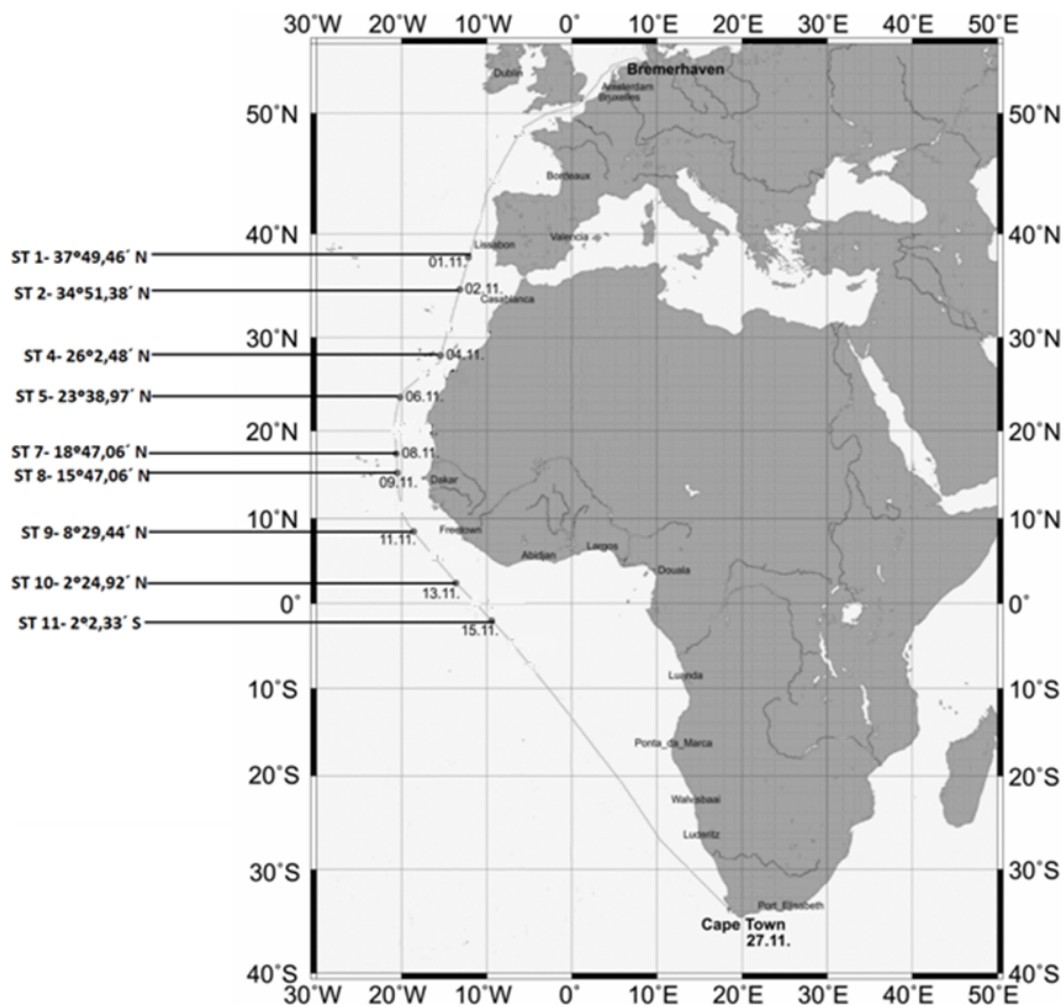

**Figure 1** Map showing the sampling stations along the Polarstern travel.

The samples from the net allocated to visual identification were divided at random (haphazardly) in two parts. One was employed for other purposes (teaching project within the cruise EUROPA), and the other was examined under binocular microscope. Fish were identified *de visu*, sorted manually and stored in 96% ethanol until genetic analysis. A few individuals were taxonomically classified to the lowest taxonomic level using a stereomicroscope (Leika MS5) and identification guides (*Olivar & Fortuno, 1991*; *Boltovskoy, 1999*; *Richards, 2005*) for comparison of visual identification with DNA barcoding methodology. Insights on diagnostic morphological traits from the online database FishBase (*Froese & Pauly, 2016*) were also used.

Abundances for the different groups were calculated in individuals per 1,000 m³. The volume of water filtered by the net was estimated using a flow meter (Hydro bios) attached to one Bongo net.

The locations and coordinates of the sampling sites are provided in Fig. 1.

## DNA barcoding

Total DNA was extracted from a small piece of tissue following the standard protocol described by *Estoup et al. (1996)*, employing Chelex® resin (Bio-Rad Laboratories). The tubes were stored at 4 °C for immediate DNA analysis, and aliquots were frozen at −20 °C for longtime preservation.

A fragment of the COI gene was amplified by polymerase chain reaction (PCR), employing the primers described by *Ward et al. (2005)*. The amplification reaction was performed in a total volume of 40 µl, with Promega (Madison, WI) buffer 1X, 2.5 mM MgCl$_2$, 0.25 mM dNTPs, 20 pmol of each primer, approximately 20 ng of template DNA and 1 U of DNA Taq polymerase (Promega), and the following PCR conditions: initial denaturing at 95 °C for 5 min, 35 cycles of denaturing at 95 °C for 20 s, annealing at 57 °C for 20 s, extension at 72 °C for 30 s and final extension at 72 °C for 10 min.

Sequencing was performed with the DNA sequencing service Macrogen Europe (Amsterdam,The Netherlands).

## Species identification based on DNA

Sequences were visualized and edited with the programme BioEdit Sequence Alignment Editor (*Hall, 1999*) and aligned with ClustalW (*Thompson, Higgins & Gibson, 1994*) as implemented in BioEdit. First the sequences were checked for the possibility of being pseudogenes (*Bensasson et al., 2001*; *Richly & Leister, 2004*) using the putative translated protein as a quality filter, since pseudogenes rarely conserve the reading framework of true coding genes. The putative protein was inferred from the amplicons using MEGA 6 software (*Tamura et al., 2013*). For identifying the species, the sequences obtained were compared with international databases using the BOLD system (http://www.boldsystems.org/), based on the Hidden Markov Model (HMM) algorithm (*Eddy, 1998*) and the BLAST algorithm of GenBank (*Altschul et al., 1990*) (http://www.ncbi.nlm.nih.gov/). Besides, a Neighbour –Joining (NJ) tree has been built with a distance-based approach to illustrate the sequence identity based on tree topology. The phylogenetic analysis was performed using MEGA 6 (*Tamura et al., 2013*).

## RESULTS

DNA extraction and PCR amplification of the COI gene was successful in 90% of the samples. The 237 sequences obtained had an average length of 630 nucleotides. Pseudogenes were reasonably discarded since putative proteins obtained from translation of the amplicons were all compatible with coding COI genes (*Bensasson et al., 2001*; *Richly & Leister, 2004*). Not all of them retrieved a significant match at species level from the reference databases (Table 1), with the cutoffs used in this study (>97% identity, >80% coverage) within the 97–97.4% commonly accepted for this gene in Barcoding projects (e.g., *Meyer & Paulay, 2005*). In some cases, species identification was not possible because two or more reference sequences of different species in BOLD were identical, sometimes even from different families (Table 1). These individuals were classified as ambiguities, because it was not possible to identify them using GenBank. In other cases there was clear discrepancy between the two databases, retrieving different species, and sometimes

**Table 1** **Assignment of COI gene sequences of fish larvae collected along a latitudinal gradient in the Eastern Atlantic Ocean to taxonomic groups in the databases GenBank and BOLD.** Number of sequences assigned to only one of the databases, to the two of them (concordance), discrepancy between the two databases (discrepancy), and ambiguous assignment due to identical match in BOLD (ambiguity), and number of sequences assigned to each taxonomic level (final assignment). The sequences finally assigned to a level include the concordant cases in such level plus the discrepant/ambiguous cases of the lower level.

|  | GenBank | BOLD | Concordance | Discrepancy | Ambiguity | Final Assignment |
|---|---|---|---|---|---|---|
| Order | 8 | 0 | 0 | 0 | 0 | 10 |
| Family | 0 | 0 | 0 | 0 | 2 | 22 |
| Genus | 0 | 5 | 11 | 4 | 18 | 84 |
| Species | 0 | 58 | 63 | 2 | 66 | 121 |
| $n$ | 8 | 63 | 74 | 6 | 86 | 237 |

even different genera. In cases of discrepancy, the individual was assigned to the higher taxonomic level (Table 1). The sequences that were identified at species level in BOLD and/or by BLAST in (44 species) were submitted to GenBank (Table 2). The phylogenetic classification of these 44 species was checked with a Neighbour-Joining tree methodology (Fig. 2). From the 237 individuals barcoded, 121 (51.1%) were assigned to a species, and 205 (86.5%) to a genus (Table 1). All the individuals were assigned to an order and most of them to a family (96.6%). Finally, a Neighbour-Joining (NJ) tree was built with a distance-based approach to illustrate the sequence identity based on tree topology (Fig. 2). The trees obtained from the two approaches were largely consistent.

The taxonomic resolution, obtained from the COI gene and the databases used as references, was different for the eight orders found in this study (Table 2). In other words, we found differences between groups of fish for their coverage within the databases employed as references. In the Myctophiformes ($n = 164$) only 59% of the individuals were identified to a species level (Fig. 3). For the less abundant Stomiiformes ($n = 32$) and Perciformes ($n = 21$), only 28% and 19 % could be identified to a species level, respectively. Since the orders were not equally abundant at all stations, and the specific resolution of the orders was different, as a result the taxonomic resolution was spatially different along the latitudinal gradient considered. The number and proportion of individuals identified to a species level was higher at the stations at higher latitudes and decreased towards the equator (Fig. 4).

Using five species of Myctophiformes (Table 2), the comparison between morphological and DNA-based species identification was done taking individuals at random from the list. These were *Diogenichthys atlanticus* (3 individuals in our sample), *Hygophum hygomii* (22 individuals), *Lampanyctus alatus* (1 individual), *Lampanyctus pusillus* (3 individuals) and *Myctophum asperum* (1 individual). Inconsistencies between visual taxonomic and barcoding identification were not found. The time required for identifying one individual by a researcher or technician will vary, depending on previous training and familiarity with the species and the accurate identification must be based on diagnostic characters (Table 3). Here, the identification time was estimated from the approximate time required by a trained technician to check the diagnostic traits indicated in Table 3 under the

**Table 2** Identified fish larvae along a latitudinal gradient in the Eastern Atlantic Ocean based on COI sequences.

| Order | Family | Genus | Species | n | St | GenBank |
|---|---|---|---|---|---|---|
| Anguilliformes | Congridae | *Conger* | *Conger conger* | 1 | 1 | KU902905 |
| | Ophichthidae | *Dalophis* | *Dalophis imberbis* | 1 | 1 | KU902930 |
| | | *Myrophis* | *Myrophis punctatus* | 1 | 1 | KU902940 |
| Argentiniformes | Microstomatidae | *Batylagoides* | *Batylagoides argyrogaster* | 2 | 1 | KU902900 |
| Aulopiformes | Evermannellidae | *Evermanella* | *Evermanella balbo* | 1 | 1 | KU902909 |
| | Paralepididae | *Macroparalepis* | *Macroparalepis affinis* | 1 | 1 | KU902936 |
| | Scopelarchidae | *Benthabella infans* | *Benthabella infans* | 1 | 1 | KU902901 |
| Lophiiformes | Melanocetidae | *Melanocetus* | *Melanocetus johnsonii* | 1 | 1 | KU902937 |
| Myctophiformes | Myctophidae | *Benthosema* | *Benthosema suborbitale* | 4 | 1 | KU902902 |
| | | *Bolinichthys* | *Bolinichthys indicus* | 5 | 3 | KU902903 |
| | | *Ceratoscopelus* | *Ceratoscopelus maderensis* | 1 | 1 | KU902904 |
| | | *Diaphus* | *Diaphus brachycephalus* | 1 | 1 | KU902931 |
| | | | *Diaphus holti* | 1 | 1 | KU902907 |
| | | | *Diaphus rafinesquii* | 8 | 2 | KU902908 |
| | | *Diogenichthys* | *Diogenichthys atlanticus* | 3 | 2 | KU902932 |
| | | *Hygophum* | *Hygophum hygomii* | 22 | 3 | KU902912 |
| | | | *Hygophum macrochir* | 24 | 5 | KU902913 |
| | | | *Hygophum taaningi* | 1 | 1 | KU902914 |
| | | *Lampadena* | *Lampadena pontifex* | 1 | 1 | KU902915 |
| | | *Lampanyctus* | *Lampanyctus alatus* | 1 | 1 | KU902934 |
| | | | *Lampanyctus nobilis* | 1 | 1 | KU902916 |
| | | | *Lampanyctus pusillus* | 3 | 2 | KU902917 |
| | | *Lepidophanes* | *Lepidophanes guentheri* | 4 | 1 | KU902935 |
| | | *Lobianchia* | *Lobianchia dofleini* | 1 | 1 | KU902918 |
| | | *Myctophum* | *Myctophum affine* | 1 | 1 | KU902920 |
| | | | *Myctophum asperum* | 1 | 1 | KU902921 |
| | | | *Myctophum obtusirostre* | 1 | 1 | KU902938 |
| | | | *Myctophum selenops* | 2 | 1 | KU902939 |
| | | *Notolychrus* | *Notolychrus valdiviae* | 1 | 1 | KU902941 |
| | | *Notoscopelus* | *Notoscopelus resplendens* | 5 | 2 | KU902922 |
| | | *Symbolophorus* | *Symbolophorus veranyi* | 1 | 1 | KU902925 |
| | | *Taanngichthys* | *Taanngichthys minimus* | 1 | 1 | KU902942 |
| Pleuronectiformes | Pleuronectidae | *Reinhardtius* | *Reinhardtius hippoglossoides* | 2 | 2 | KU902924 |
| Scombriformes | Gempylidae | *Diplospinus* | *Diplospinus multistriatus* | 2 | 1 | KU902933 |
| Scorpaeniformes | Serranidae | *Mycteroperca* | *Mycteroperca acutirostris* | 1 | 1 | KU902919 |
| Stomiatiformes | Gonostomatidae | *Cyclothone* | *Cyclothone acclinidens* | 1 | 1 | KU902928 |
| | | | *Cyclothone braueri* | 1 | 1 | KU902906 |
| | | | *Cyclothone livida* | 1 | 1 | KU902929 |
| | | *Gonostoma* | *Gonostoma denudatum* | 1 | 1 | KU902910 |
| | | | *Gonostoma elongatum* | 2 | 1 | KU902911 |
| | Stomiidae | *Astronethes* | *Astronethes richardsonii* | 1 | 1 | KU902899 |

**Table 2** (*continued*)

| Order | Family | Genus | Species | *n* | *St* | GenBank |
|-------|--------|-------|---------|-----|------|---------|
| Stomiiformes | Phosichthyidae | *Vinciguerria* | *Vinciguerria nimbaria* | 2 | 2 | KU902926 |
| Stromateiformes | Nomeidae | *Cubiceps* | *Cubiceps baxteri* | 1 | 1 | KU902927 |
| | | *Psenes* | *Psenes arafurensis* | 2 | 2 | KU902923 |

**Notes.**
    *n*, Number of individuals of each species; *St*, Number of stations where a species occurred; GenBank, Accession numbers of the COI sequences.

microscope or magnifying glass (depending on the size of the individual). Less time is needed if the species has a distinctive morphology within the order (big or small eyes, atypical shape, unusual characters). The main point here is that the individuals need to be examined one by one for visual classification. For DNA barcoding, the time dedicated to each individual is shorter: just cutting a piece of tissue and storing it in ethanol. The process of labeling vials with adequate codes for traceability between the tissue sample and the voucher specimen can be very fast with pre-prepared labels. An experienced technician can complete DNA extraction and preparation of PCR quite rapidly (Table 4). These processes can be robotized, thus a large number of samples can be analyzed in relatively short time. In the present case, samples of $N = 30$ from three species were loaded simultaneously in a thermocycler of 96 wells. Briefly, more labor hours and cheaper analytical materials are required for visual analysis than for DNA barcoding. We have estimated time costs per hour using the average salary per working hour of a Spanish laboratory technician in 2015, because the laboratory analysis was done in Spain. The final cost of analyzing the 30 individuals of those five species found in our sampling was €169 and €90.5 for DNA barcoding and visual identification, respectively.

Most of the species were only collected at one station and only three myctophid species were found at more than two stations: *Bolinichthys indicus*, *Hygophum hygomii* and *H. macrochir* (Table 2).

## DISCUSSION

The main shortcoming of DNA barcoding in our study was the lack of a solid, large and robust database with reliable reference sequences. The same problem was found by *Leis (2015)* for Indo-Pacific fish larvae. DNA helps to establish identities of larvae when there are reference barcodes with accompanying voucher specimen taxonomically ascertained in the databases. This is not the case of a large proportion of the larvae found in plankton surveys. Therefore the enormous potential for DNA to advance larvae identification has not been properly exploited yet. An integrative approach combining genetics and diagnostic morphological traits for species identification would be strongly recommended (*Leis, 2015*). From our results, Atlantic pelagic fish identification through sequence similarity seems to be hampered by the incompleteness of the taxonomic coverage, as it was also found for Antarctic bony fish (*Dettai et al., 2011*). In addition to database incompleteness, we have found some inconsistencies in BOLD due to identical sequences assigned to different species. In this sense, generating a critical mass of BARCODE compliant specimen records and developing an error-free searchable database have been identified

Ardura et al. (2016), *PeerJ*, DOI 10.7717/peerj.2438

**Table 3** Morphological diagnostic traits for identification of fish larvae and estimated time for identification of one individual.

| | Diagnosis time | Depth range | Max length | Soft rays | | | Gill rakers | | Other diagnostic traits |
| | | | | Dorsal | Anal | Pectoral | Lower limb | Upper limb | |
|---|---|---|---|---|---|---|---|---|---|
| *Diogenichthys atlanticus* | 15 min | 0–1,050 | 2.9 | 10-12 | 15-18 | 12-14 | 10-13 | 2-2 | |
| *Hygophum hygomii* | 10 min | 0–1,485 | 6.8 | 13-15 | 21-23 | 14-17 | 14-17 | 5-6 | Big characteristic eyes |
| *Lampanyctus alatus* | 15 min | 40–1,500 | 6.1 | 11-13 | 16-18 | 11-13 | 8-10 | 2-4 | |
| *Lampanyctus pusillus* | 15 min | 40–850 | 4.3 | 11-13 | 13-15 | 13-14 | 8-10 | 3-3 | |
| *Myctophum asperum* | 10 min | 244–1,948 | 8.5 | 12-14 | 17-20 | 12-16 | 10-12 | 3-5 | Anal organs |

Ardura et al. (2016), *PeerJ*, DOI 10.7717/peerj.2438

**Table 4  Total cost estimation for the identification of the 30 individuals of the five species considered (*n* = number of individuals of each species) found in the sampling stations, based on the time required for the analysis (at left) and consumables/external sequencing (right).** Spanish salaries for laboratory technicians were taken from the official Resolution 2,000 BOE 49 of 26 of February of 2015.

| | *n* | Diagnosis time (min labour) | | | | Consumables & external analysis | |
| --- | --- | --- | --- | --- | --- | --- | --- |
| | | Visual | DNA barcoding | | | Visual | DNA barcoding |
| | | | Tissue sampling | DNA extraction | PCR preparation | | |
| *Diogenichthys atlanticus* | 3 | 15 min × 3 = 45 | 2 min × 3 = 6 min | | | Fixative | Extraction kit |
| *Hygophum hygomii* | 22 | 10 min × 22 = 220 | 2 min × 22 = 44 min | | | | PCR products |
| *Lampanyctus alatus* | 1 | 15 min × 1 = 15 | 2 min × 1 = 2 min | 15 min | 30 min | | Sequencing & BLAST |
| *Lampanyctus pusillus* | 3 | 15 min × 3 = 45 | 2 min × 3 = 6 min | | | | |
| *Myctophum asperum* | 1 | 10 min × 1 = 10 | 2 min × 1 = 2 min | | | | |
| Total time | | 335 min | 105 min | | | | |
| **Estimated cost** | | **60.5€** | **18.97€** | | | **1€×30 = 30€** | **5€×30 = 150€** |

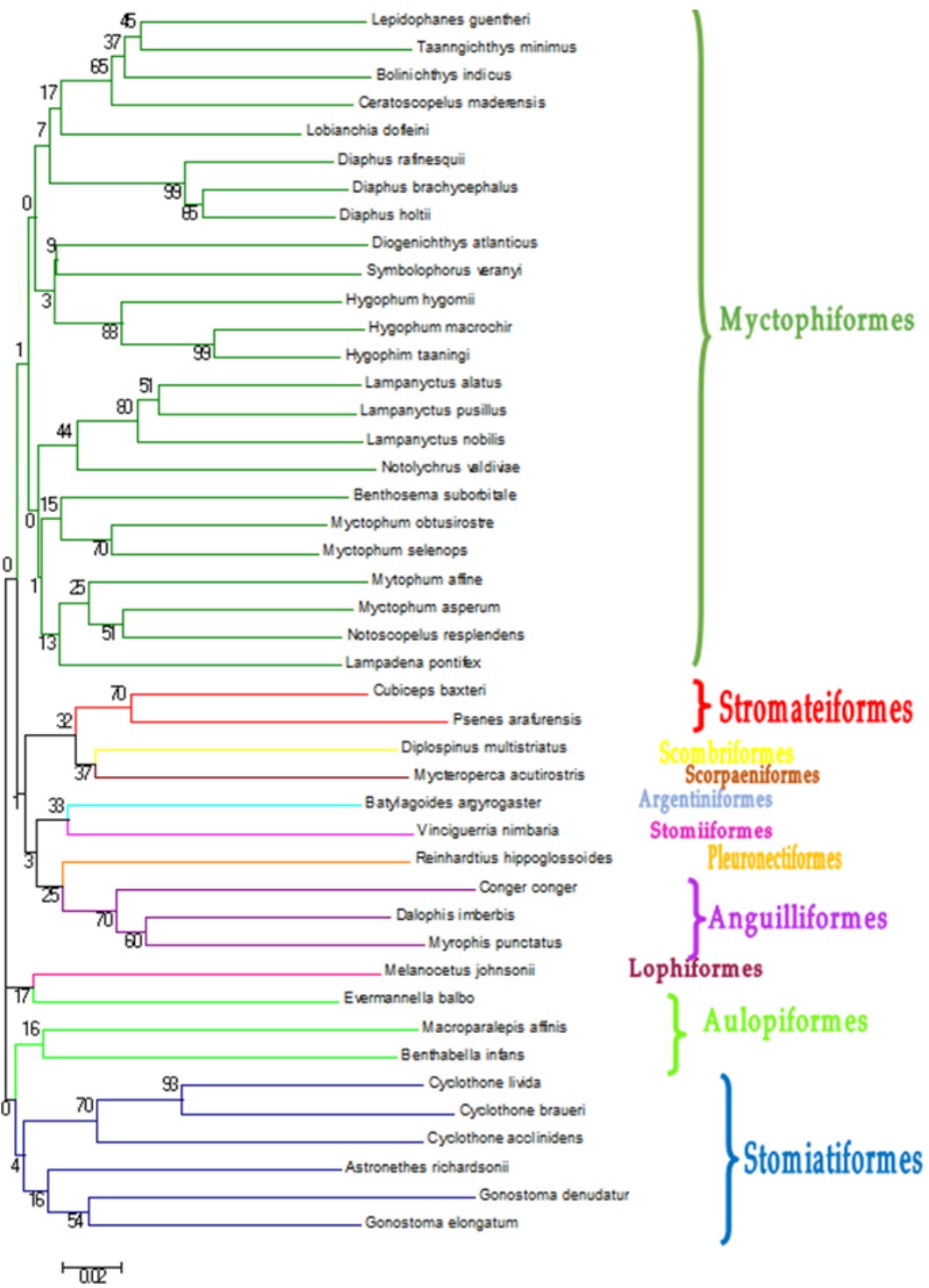

**Figure 2  Neighbour-Joining tree constructed based on COI gene haplotypes decribed in Table 2.**
Bootstrap values are presented in percent. Branches containing the different orders found in this study.
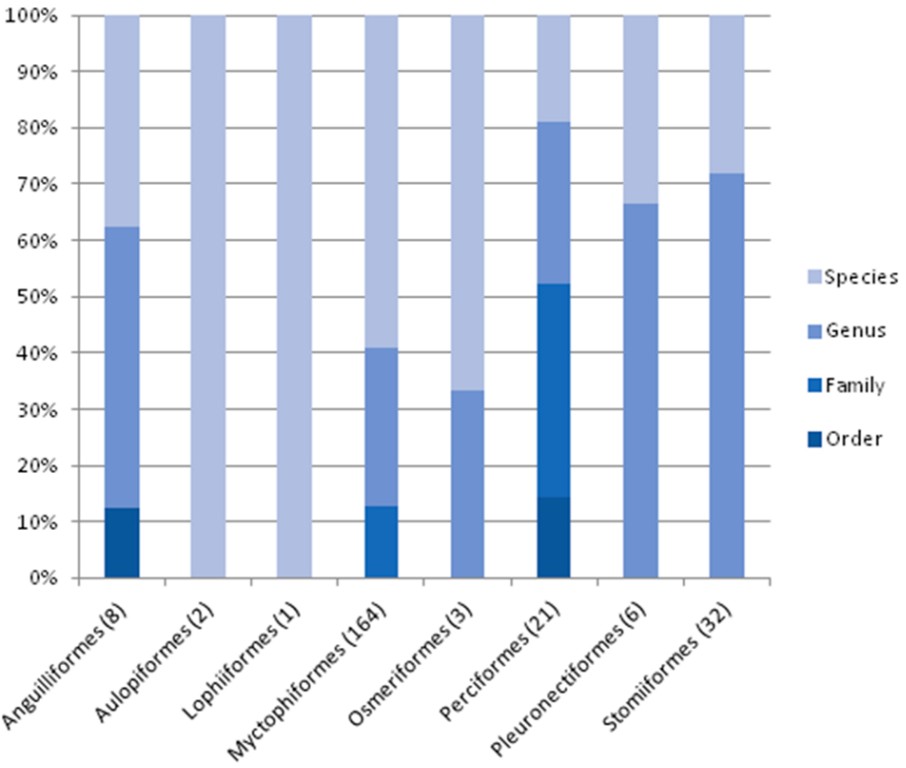

**Figure 3 Level of taxonomic resolution of the planktonic fish analyzed obtained from COI gene sequences and nBLAST methodology, for each order.** Sample size in parenthesis.

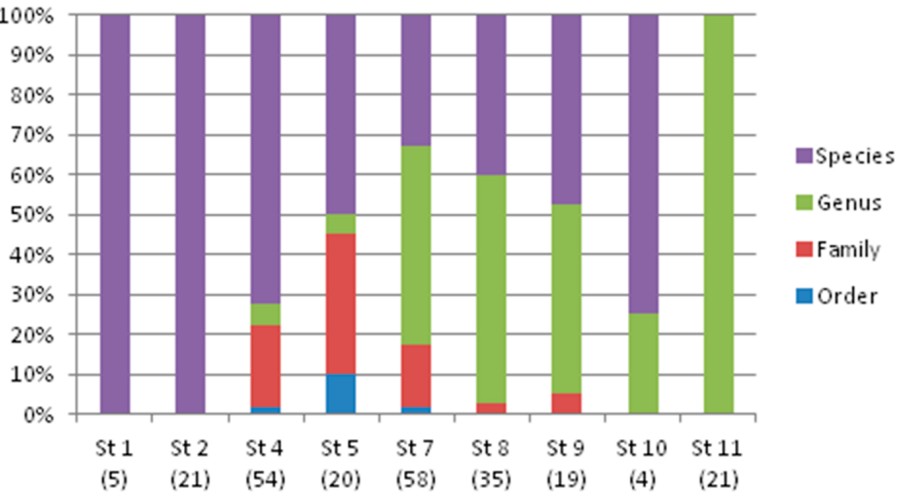

**Figure 4 Taxonomic resolution of the individuals sampled from each station (*St*), obtained from DNA barcoding.** Sample size in parenthesis.

as critically important issues for FISH-BOL to tackle (*Ward, Hanner & Hebert, 2009*). For large scale surveys of planktonic fish, those inconsistencies should be solved if barcoding is to be applied. Recommendations could be, for example, curating the databases against voucher specimens and improving reference collections agreed among experts from different countries and disciplines.

In our study, a clear economic advantage has been found for visual methodology using a short subset of samples (Table 4). However, if we apply these estimates to a larger dataset the difference between methods will decrease, or turn to be advantageous for DNA-based methodologies because they can be robotized and time saved. However, equipment is objectively more expensive for DNA analysis than for visual methodology today. Cost per individual decreases are predicted for genomic methods, thus DNA barcoding-based approach is likely to become cost-effective in the near future (*Edwards et al., 2010*).

Finally, Myctophiformes (lanternfish) and Stomiiformes (dragonfish), the most abundant taxa and lacking accurate references for being identified to the species level (Fig. 3), are the critical trophic link between primary producers and higher trophic levels including important fishing resources such as tuna (e.g., *Choy et al., 2012* and references therein). Ecosystem-based management of fishing resources (e.g., *Hall & Mainprize, 2004*) would be greatly improved if diversity estimates of these important plankton components are accurate, and enhancing barcoding databases could contribute efficiently to this.

## CONCLUSIONS

A lack of accurate unambiguous DNA sequence references was found for Atlantic bathypelagic fish, especially in tropical latitudes. These results emphasize the need of obtaining more genetic Barcodes for bathypelagic fish species to increase the potential utility of barcoding for diversity estimates in ecosystem-based management of fishing resources.

## ACKNOWLEDGEMENTS

We would like to thank the institutions and persons that made our work possible: Alfred Wegener Institute for Polar and Marine Research (Bremerhaven, Germany); crew of RV Polarstern (ANT-XXIX/1); Holger Auel (University of Bremen, Germany), for coordinating the expedition.

### Funding

Clarin-COFUND-PCTI-FICYT provided a post-doc fellowship to AA. The funders had no role in study design, data collection and analysis, decision to publish, or preparation of the manuscript.

### Grant Disclosures

The following grant information was disclosed by the authors:
Clarin-COFUND-PCTI-FICYT.

## Competing Interests

The authors declare there are no competing interests.

## Author Contributions

- Alba Ardura and Eva Garcia-Vazquez conceived and designed the experiments, performed the experiments, analyzed the data, contributed reagents/materials/analysis tools, wrote the paper, prepared figures and/or tables.
- Elvira Morote and Marc Kochzius contributed reagents/materials/analysis tools, reviewed drafts of the paper.

## DNA Deposition

The following information was supplied regarding the deposition of DNA sequences:
GenBank, KU902899–KU902942.

## Supplemental Information

Supplemental information for this article can be found online at http://dx.doi.org/10.7717/peerj.2438#supplemental-information.

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
