# Peer review of "Diversity of planktonic fish larvae along a latitudinal gradient in the Eastern Atlantic Ocean estimated through DNA barcodes"

_PeerJ, doi:10.7717/peerj.2438_

## Round 0.1 · original submission · Major Revisions

Please, consider all the suggestions in the revised manuscript.

Reviewer 1 ·

Basic reporting

This manuscript presents interesting and important results on ichtioplankton detected by DNA barcoding during the cross-latitudinal expedition in Eastern Atlantic, along with some insights into the comparison of visual identification with barcoding approach.

In general, the text is well written, but some parts of the manuscript require additional English check, as do not read well (particularly, the Discussion section).

The structure of the manuscript complies with the PeerJ requirements.

I have some comments on the Figure 1 - there are many additional dots on the route line that are nor explained in legend neither addressed in the text. Were these sampling stations as well? If so, why no results are provided for them? If not, why they are shown? they introduce unnecessary noise and complicate the readability.

Experimental design

The submission presents an original primary research and the primary research question of this study was "to identify current needs for the full application of DNA barcoding to the inventory of planktonic fish in large scale surveys." The applied methods were relevant for this objective, however description of sampling strategy and samples' analysis could be clarified better.

For example, it is not quite clear how the samples were split and which were used for comparative analysis.
Lines 73-87: Why samples from separate Bongo nets were split first for visual and barcoding identification (I suggest to spell barcoding rather than Barcoding throughout the text), and then divided into two parts again (Line 80). Please explain better this part.

Validity of the findings

Although the collected data look robust, and results are scientifically sound, corresponding to the identified objective, I have some comments on some of the conclusions and statements made by authors.
First of all, I'm not convinced that the collected dataset is suitable for the formal assessment of biodiversity. I think the sampling effort was not sufficient to make any inferences on the ichtyoplankton species richness, evenness and taxonomic diversity (actually, I guess it was taxonomic distinctness calculated in Primer). The calculated indices look weird and they are not addressed in discussion, so their assessment is not justified at all. I would suggest to exclude these metrics from the manuscript, since they do not have any added value, but rather rise many questions about their validity.
On the other hand, some of the findings relevant to the primary research question are not well covered in the discussion. For example:
why taxonomic resolution decreases at lower latitudes?
why there were inconsistencies in database records for one species but not the other?
what are the implications of the current findings for the large scale surveys of planktonic fish?
were there any inconsistencies between visual identifications and barcoding results? If so, what are the reasons?
etc.

On the other hand, some of the statements are quite speculative and not well supported either by current findings or literature cited. For example:
Lines 153-154: how the overlapping distributions were assessed? did the authors sampled different water layers? or formally analyzed co-occurence of the taxa from different latitudes? if so, this should be clearly described in Methods, otherwise they need to describe how it was done (based on literature, previous studies, etc.)
Lines 205-206: this statement is not supported by results. Is there any study showing how the cost of barcoding decreasing with the number of samples? If this is speculative statement, if should be clearly said, that this is an assumption, but not the proven fact.
Line 216: I couldn't find any information on the oxygen concentration and report of oxygen depletion zones in this study. So this statement should be also reasoned by literature or other results.

Additional comments

Some specific comments on the text:

Line 91: was it a single location or locationS?
Line 117: this is redundant, omit.
Line 119: Copyright for Primer should be indicated.
Line 120: in a habitat
Lines 119-122: does not read well, correct please.
Lines 136-137: "sequences able to identify" sounds weird, rephrase please
Lines 144, 146, 149: to a species level
Lines 153-154: The sentence does not read well, rephrase
Line 171: In summary, more...
Lines 170-176: revise English
Lines 180-183: how can sample size increase by switching from species to genus level? Clarify what do you mean by "sample size" (also in Table 2), or rephrase, since this does not make sense
Lines 178-202: does not read well, revise English
Lines 194-196: these are results, not discussion
Lines 198-201: rephrase, does not read well
Line 204: "visual methodology", "visual observation" - does it stand for visual identification? Please be consistent with wording throughout the text.

Reviewer 2 ·

Basic reporting

The professional English language used need to be revised. I have noticed some grammar mistakes .
The context is well shown in the introduction and literature is well referenced.
Structure conforms to PeerJ standard. However, the authors should check the reference list following the PeerJ guidelines.
Figures are high quality and well labeled and described.

Experimental design

This is an original primary research within the Biological Science and then within scope of Peer J.
The research question is meaningful dealing with diversity of mid-trophic planktonic fish larvae along the Eastern Atlantic Ocean that is a very important area for fisheries. Further, the biodiversity of fish larvae is a poorly studied research area.
The investigation was performed rigorously

Validity of the findings

Data is robust but need to be better analyzed. I suggest to implement genetic analyses performing most common used approaches. Generally, two approaches were employed to analyze DNA barcode sequences and to verify the identity of unknown samples: a similarity search which is conducted with the DNA Identification Engine at BOLD (Barcode of Life Database), based on the Hidden Markov Model (HMM) algorithm (Eddy 1998), and BLAST algorithm of GenBank (Altschul et al. 1990); and Neighbour-Joining (NJ) trees built with a distance-based approach to illustrate sequence identity based on tree topology.
The conclusions were connected to the original question investigated.

Additional comments

The manuscript” Diversity of plantonic fish larvae along a longitudinal gradient in the Eastern Atlantic Ocean estimated through DNA barcodes” by Ardura et al. analyzed the biodiversity of planktonic fish larvae using a fragment of the COI gene sequence (as DNA barcode).
The molecular data need to be better analyzed. I have noticed some grammar mistakes . I recommend the English revision of the text.
All my comments and suggestions are enclosed below. I hope they will be welcomed and useful for the authors.
I suggest a few points that should be improved:

Introduction

Line 56: Recently a paper on the species identification of Mediterranean lanternfish larvae by COI DNA barcoding was published. I suggest to insert this reference.“ Pappalardo A.M., Cuttitta A., Sardella A., Musco M., Maggio T., Patti B., Mazzola S., Ferrito V. (2015) DNA Barcoding and COI sequence variation in Mediterranean lanternfishes larvae. Hydrobiologia 745: 155-167”.
Line 40: Please replace “Walker, Mead and Brownell, 2002” by “ Walker, Mead & Brownell, 2002”.
Line 42: Please delete “,” .

Materials and methods
Due to the aim of this work I suggest to implement genetic analyses performing most common used approaches. Generally, two approaches were employed to analyze DNA barcode sequences and to verify the identity of unknown samples: a similarity search which is conducted with the DNA Identification Engine at BOLD (Barcode of Life Database), based on the Hidden Markov Model (HMM) algorithm (Eddy 1998), and BLAST algorithm of GenBank (Altschul et al. 1990); and Neighbour-Joining (NJ) trees built with a distance-based approach to illustrate sequence identity based on tree topology.
Have the authors investigated on the presence of NUMTs (transfers of mtDNA COI sequences into the nuclear genome)? Several authors (e.g. Bensasson et al. 2001; Richly & Leister 2004) confirms the need for vigilance in examining fish amplicons for potential pseudogene status.



Minor comments:
Line 84: Please replace “Oliver and Fortuno, 1991” by “Oliver & Fortuno, 1991”.
Line 85: Please make uniform throught the text “DNA barcoding” or “DNA Barcoding”
Line 99: Add the buffer concentration.

Results
Line 130: the cutoffs used for species identification (>90% identity) is too low. Generally, samples were considered identified at species level when there was less than 1% difference with reference sequences (see Ratnasingham and Hebert, 2007).

References
The authors should check the reference list following the PeerJ guidelines.
The paper “Boltovskoy, 1999” is not cited in the Reference list.

---

## Round 0.2 · accepted · Accept

Your manuscript has been improved and is now ready for publication,

Reviewer 2 ·

Basic reporting

The article is well written in English.

Experimental design

The manuscripts describe original primary research within the scope of the PeerJ.

Validity of the findings

The data are robust and statistically sound.

Additional comments

The new revision of this manuscript is much improved and the authors have done a satisfactory job of addressing the reviewer comments.